# Creation of a biological sensorimotor interface for bionic reconstruction

Christopher Festin [1,2], Joachim Ortmayr[1], Udo Maierhofer [1,2],
Vlad Tereshenko [1,2,3], Roland Blumer [4], Martin Schmoll[5],
Génova Carrero-Rojas[4], Matthias Luft[1,2,6], Gregor Laengle [1,2,7], Dario Farina [8],
Konstantin D. Bergmeister[1,6] & Oskar C. Aszmann [1,7] ✉

Neuromuscular control of bionic arms has constantly improved over the past years, however, restoration of sensation remains elusive. Previous approaches to reestablish sensory feedback include tactile, electrical, and peripheral nerve stimulation, however, they cannot recreate natural, intuitive sensations. Here, we establish an experimental biological sensorimotor interface and demonstrate its potential use in neuroprosthetics. We transfer a mixed nerve to a skeletal muscle combined with glabrous dermal skin transplantation, thus forming a bi-directional communication unit in a rat model. Morphological analyses indicate reinnervation of the skin, mechanoreceptors, NMJs, and muscle spindles. Furthermore, sequential retrograde labeling reveals specific sensory reinnervation at the level of the dorsal root ganglia. Electrophysiological recordings show reproducible afferent signals upon tactile stimulation and tendon manipulation. The results demonstrate the possibility of surgically creating an interface for both decoding efferent motor control, as well as encoding afferent tactile and proprioceptive feedback, and may indicate the way forward regarding clinical translation of biological communication pathways for neuroprosthetic applications.

Bionic reconstruction is an established concept for restoring hand function in cases of traumatic amputation, malignancies, severe soft tissue, or brachial plexus injuries where biological reconstruction is no longer possible[1,2]. Surgical techniques, such as targeted muscle reinnervation (TMR), osseointegration, or regenerative peripheral nerve interfaces (RPNI), as well as technological innovations, such as implantable electromyography electrodes, multi-electrode arrays, and machine learning, have greatly improved prosthetic control in recent years[3–7]. Ultimately, however, the development of clinically viable man-machine interfaces has not been able to keep pace with the increasing complexity of robotic limbs, thus creating a mismatch between the capabilities of the prosthetic end effector and the information transfer from and to the patient's nervous system. Interestingly, despite the high degree of technological sophistication, myoelectric prosthetic devices have only recently surpassed conventional body-powered prostheses in terms of popularity among patients[8,9]. Among the main reasons for the success of body-powered devices are the straightforward control, as well as instant force and proprioceptive feedback

[1]Clinical Laboratory for Bionic Extremity Reconstruction, Department of Plastic, Reconstructive and Aesthetic Surgery, Medical University of Vienna, Vienna, Austria. [2]Center for Biomedical Research, Medical University of Vienna, Vienna, Austria. [3]Division of Plastic and Reconstructive Surgery, Massachusetts General Hospital, Harvard Medical School, Boston, MA, USA. [4]Center for Anatomy and Cell Biology, Medical University of Vienna, Vienna, Austria. [5]Center for Medical Physics and Biomedical Engineering, Medical University of Vienna, Vienna, Austria. [6]Department of Plastic, Aesthetic and Reconstructive Surgery, University Hospital St. Poelten, Karl Landsteiner University of Health Sciences, Krems, Austria. [7]Department of Plastic, Reconstructive and Aesthetic Surgery, Medical University of Vienna, Vienna, Austria. [8]Department of Bioengineering, Imperial College London, London, United Kingdom. ✉e-mail: oskar.aszmann@meduniwien.ac.at

transmitted via the cables and harness[10,11]. Conversely, users of commercial myoelectric devices rely on unintuitive channels of sensation, such as pressure from the socket, proprioceptive organs of residual muscles, or auditive and visual cues from the prosthesis itself[12,13]. Therefore, the restoration of intuitive sensory feedback in bi-directional man-machine interfaces remains an unsolved challenge[12].

Due to the inadequacies of these incidental rather than intended forms of afferent information, restoring the sensation of the missing limb remains a pressing issue among patients[13,14]. For this reason, both non-invasive (sensory substitution and modality-matched feedback) and invasive (peripheral nerve stimulation) approaches have been proposed to encode sensory information and restore sensation[15,16]. Non-invasive feedback cannot usually be applied in an appropriate somatotopic manner, meaning the stimulus corresponding to sensory information, such as skin indentation, is applied to a remaining, unrelated body part (e.g., skin proximal to the amputation)[15]. While these methods do provide a certain degree of feedback, they do not replicate natural sensation, remain confined to experimental settings, and, as of now, bear low significance for long-term use in a clinical context. On the other hand, direct peripheral nerve stimulation can provide afferent stimuli. However, it is unrelated to mechanoreceptor activity, dystopic, and thus often perceived as dys- and paresthesia. There have been case studies in which proprioception was elicited, but reports in the literature are scarce and it is unclear whether current electrode technology will achieve consistent and adequate biomimetic nerve activation patterns for complex sensations[7,17–19]. It should, however, be noted that there have been reports of daily use of peripheral nerve electrodes in a small number of patients demonstrating their long-term potential, but it remains to be shown whether this concept can be translated into widespread clinical use[20,21].

The introduction of TMR and the subsequent discovery of targeted sensory reinnervation (TSR)[22] opened a new avenue for restoring sensation with a more easily accessible and intuitive interface. Essentially, the efferent fibers of a mixed nerve originally serving the missing limb reinnervate a target muscle, while the sensory fibers reinnervate the overlying skin, creating biological efferent and afferent communication pathways. Targeted reinnervation demonstrated the possibility of eliciting modality-matched, somatotopically appropriate sensations while interacting with the reinnervated skin, which provides a robust interface equipped with mechanoreceptors. It has since been employed and used in different surgical variations[23]. Combining TSR with non-invasive feedback systems may be an effective way to establish a somatotopically and biomimetically accurate sensory

experience of a patient's lost limb. Different authors have published data on utilizing transcutaneous electrical nerve stimulation[24] or mechanical pressure and vibration[25] in patients who underwent targeted reinnervation surgery with promising results. As of now, cases reported in the literature comprise the reinnervation of non-glabrous skin by nerves physiologically providing sensation to glabrous skin[22,26] and evidence suggests that the quality of the regained sensitivity is similar to that of the native skin[27]. Considering that glabrous skin additionally possesses Meissner corpuscles and a greater innervation density when compared to non-glabrous skin[28], targeted reinnervation of the former may lead to even greater restoration of sensation than previously possible. This is especially interesting in cases of elective amputation where glabrous skin, a rather scarce type of tissue to graft, could be harvested from the amputated limb and transplanted to one or more targeted muscles. The individual placement of the skin graft(s) to superficial or deeper muscles allows the creation of the most suitable neuromuscular landscape for a given amputation stump. Moreover, when using a glabrous skin graft as an encoding interface instead of the overlying non-glabrous skin, the transferred nerve fibers do not compete with native afferents, thus further supporting sensory reinnervation. Overall, it may therefore provide the best possible biological sensory interface to overcome the demanding issues of feedback regarding somatotopy and biomimicry. Here, we established a biological sensorimotor interface in a targeted reinnervation animal model utilizing a glabrous dermal skin graft transplantation. We transferred a nerve consisting of both motor and sensory axons to a denervated, but otherwise intact and in situ, skeletal muscle. Subsequently, a glabrous dermal skin graft harvested from the hindlimb was placed on top of the muscle, thus creating a muscle skin complex for bi-directional communication. Following a 12-week follow-up period, we performed immunofluorescence staining on thin and thick frozen sections, the first sequential retrograde labeling in the context of TSR to both visualize and quantify reinnervation and examined the interface's functionality using electroneurography (ENG) to record afferent nerve activity.

## Results

### Surgical feasibility

The ulnar nerve (UN), consisting of both motor and sensory axons, was transferred to the long head of the biceps (LH). A glabrous dermal skin graft was then harvested from the ipsilateral hindlimb and transplanted on top of the LH ($n = 22$, Fig. 1). Previous work highlighted the importance of a donor nerve's axonal input in motor reinnervation[29].

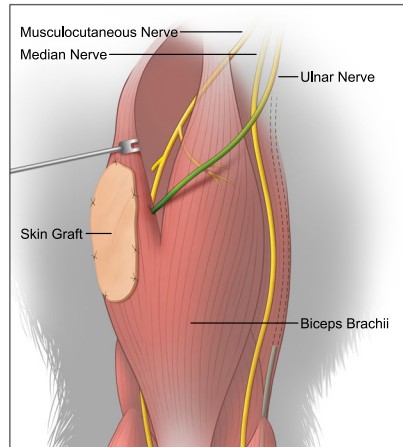
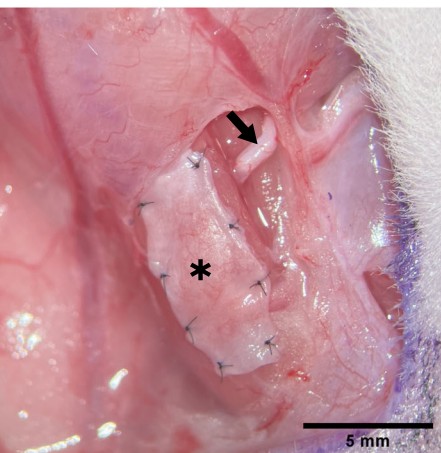
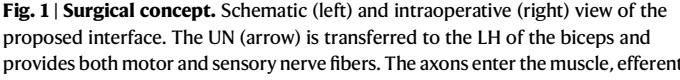

**Fig. 1 | Surgical concept.** Schematic (left) and intraoperative (right) view of the proposed interface. The UN (arrow) is transferred to the LH of the biceps and provides both motor and sensory nerve fibers. The axons enter the muscle, efferent fibers reinnervate the muscle and a portion of afferent fibers extend into the overlying glabrous dermal skin graft (*), thus creating a bi-directional, biological interface.

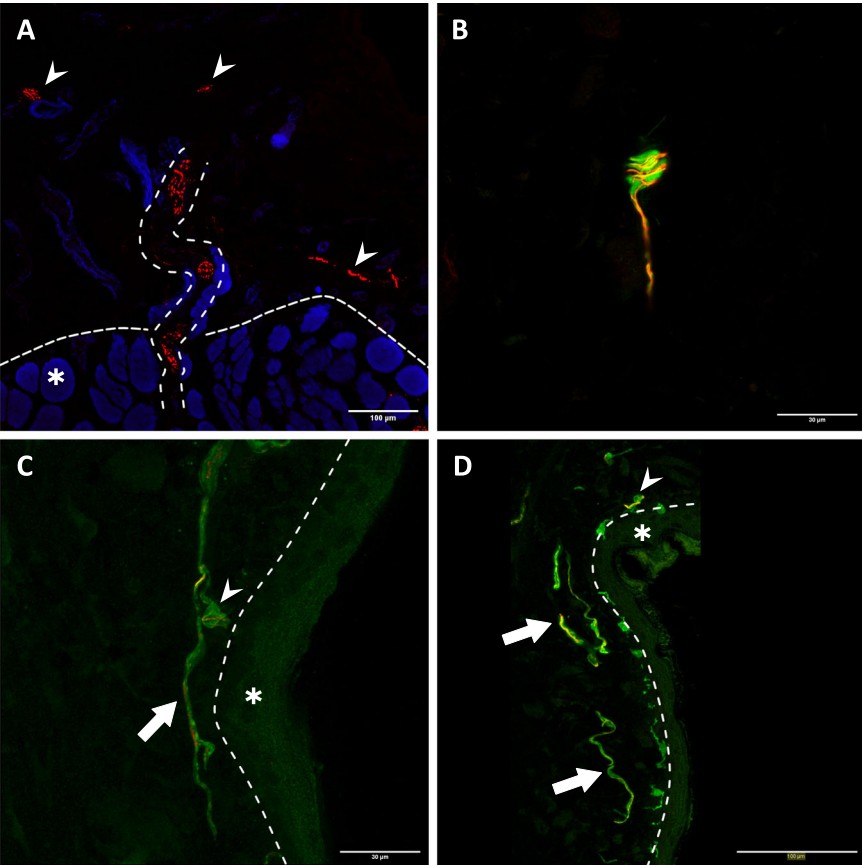

**Fig. 2 | Transmuscular reinnervation of the skin.** ($n$ = 12 for all stainings, biological replicates) **A** Anti-NF staining of the musculocutaneous junction. The densely dashed line separates the muscle fibers (*) from the dermal graft. The loosely dashed line marks a nerve fiber bundle crossing from the muscle into the overlying skin. The small arrows highlight different reinnervating nerve fibers in the skin. **B** A Meissner corpuscle in the graft with the typical lamellar morphology visualized through anti-S100 staining (green). The corpuscle is reinnervated by an NF-positive fiber (red). **C, D** Overview of the apical aspect of the transplanted skin. The larger arrows point to the re-established S100- (green) and NF-positive (red) dermal plexus. The small arrows point at Merkel disks located at the epidermal-dermal junction (dashed line) with NF-positive reinnervating nerve fibers (red). The epidermis is marked with an (*). (NF neurofilament).

For this reason, frozen sections of the native, contralateral UN, and biceps brachii's long head's muscle branch (LHMB) (each $n$ = 6) were subjected to an immunofluorescence staining protocol[30] to determine both the motor and sensory axonal load and, thus, establish reference values for future investigations. Antibodies against neurofilament (NF) act as a general neural marker and bind to all axons in a nerve cross-section, whereas choline acetyltransferase (ChAT) is a specific marker for cholinergic structures. By combining the two, it is possible to quantify all axons and distinguish between afferent (only NF-positive) and efferent (NF-/ChAT-positive) ones. Analysis of the UN and LHMB revealed 2502 ± 238 total fibers of which 309 ± 65 were also ChAT-positive and 356 ± 57 total fibers of which 156 ± 29 were also ChAT-positive, respectively (Supplemental Fig. S1A–C and Supplemental Table S1). All animals survived the surgical procedures and 12-week follow-up periods without any adverse events. Donor site morbidity was low, and all rats resumed normal gait behavior within hours after surgery. No noticeable deficits in daily activities were observed in either the affected upper or lower extremities during the entire follow-up period. Upon surgical exploration, all animals had developed epidermoid cysts from the skin graft (Supplemental Fig. S2A). We suspect that de-epithelization with a standard 15-blade did not entirely remove all epithelial cells even though great care was taken to remove all visible evidence of epidermis during the initial surgical procedure. The cysts, however, could easily be removed and did not affect further tissue analysis. Intraoperatively, the dermal skin graft appeared pink with visible signs of revascularization indicating full engraftment (Supplemental Fig. S2B). Harvested muscle-skin samples were stained with hematoxylin and eosin (H&E) demonstrating full engraftment of the entire dermal skin graft with a thin layer of epithelial cells, corresponding to the intraoperatively found epidermoid cysts, in all animals. The muscle fibers appeared intact suggesting successful motor reinnervation. Only a few individual fibers immediately at the musculocutaneous junction had a cylindrical shape, indicative of atrophy (Supplemental Fig. S3).

## Morphological evidence of targeted reinnervation

Intraoperative findings and histological analysis confirmed the model's surgical feasibility. To investigate the reinnervation of the muscle and skin by the transferred UN, 20 μm frozen muscle-skin cross sections were subjected to immunofluorescence staining with antibodies against NF, S100 and myelin basic protein (MPB) ($n$ = 13). Subsequent examination of the stained sections with fluorescence microscopy revealed the presence of NF immunoreactive nerve fibers in the dermal graft (Supplemental Fig. S3) in 12 of the 13 samples and throughout the entire muscle in all 13 samples, thus indicating successful motor and sensory reinnervation. These fibers entered the graft via transmuscular reinnervation (i.e., sensory fibers traversing and then exiting the muscle into the skin) with nerve fiber bundles passing through the musculocutaneous junction of the biological interface (Fig. 2A). Meissner corpuscle reinnervation (Fig. 2B) and the re-establishment of a dermal plexus with Merkel disks at the apex of the dermis (Fig. 2C, D) were visualized via anti-S100 staining. The use of anti-MBP antibodies demonstrated that reinnervating nerve fiber bundles were composed of mainly unmyelinated and a few myelinated fibers (Fig. 3A).

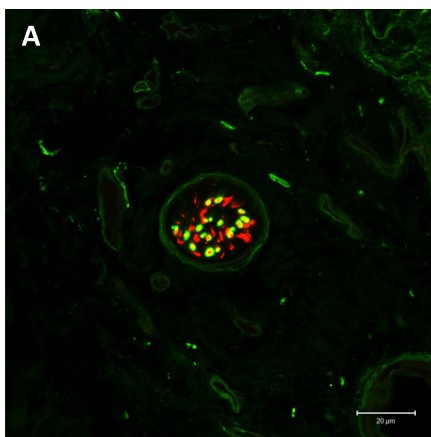
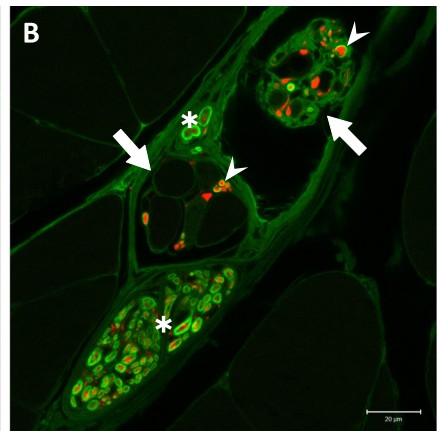

**Fig. 3 | Morphology of the reinnervating nerve fibers. A** cross-section of a nerve fiber bundle in the dermal graft. It contains many small- to medium-sized NF-positive nerve fibers (red), however, only some have a myelin sheath as visualized by the anti-MBP staining (green). ($n = 12$, biological replicates) **B** Two reinnervated muscle spindles (large arrows) within their respective capsules. They are accompanied by bundles of large, myelinated (S100, green) nerve fibers (NF, red) (*). These types of fibers can also be seen attached to the intrafusal fibers (small arrows), suggesting robust proprioceptive reinnervation. ($n = 13$, biological replicates) (NF neurofilament, MBP myelin basic protein).

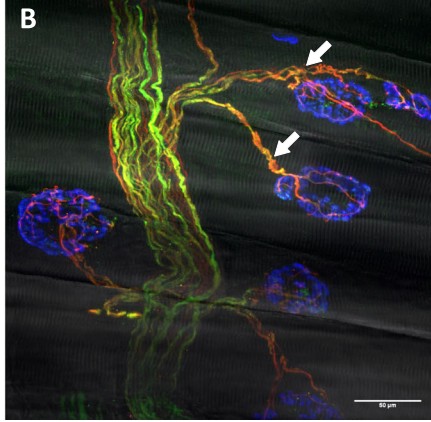

**Fig. 4 | Immunofluorescent visualization of motor reinnervation.** ($n = 8$ for all stainings, biological replicates) **A** The transferred UN (NF, red) arborizes in the LH of the biceps and reinnervates the NMJs (BTX, green) of muscle fibers (phalloidin, blue). **B** Close-up view of motor reinnervation. Nerve fibers were stained with anti-NF (red) and anti-ChAT (green) antibodies, thus visualizing motor fibers through the overlay (yellow). This confirms the specific motor reinnervation (arrows) of NMJs (BTX, blue). The muscle fibers' striation can be seen through the overlay of a brightfield image. (NF neurofilament, NMJ neuromuscular junction, BTX bungarotoxin, ChAT choline acetyltransferase).

Furthermore, it was possible to visualize muscle spindles. These spindles were accompanied by an abundance of large, myelinated nerve fibers which also wrapped around the intrafusal muscle fibers (Fig. 3B). Next to the thin frozen sections, 400 μm thick muscle-skin sections were stained to gain more morphological insight into motor and sensory reinnervation. Samples were stained with two triple-labeling antibody combinations consisting of anti-NF, bungarotoxin (BTX), and phalloidin or anti-NF, anti-ChAT, and BTX and qualitatively analyzed using confocal microscopy, similar to previous work[31]. The UN could be seen arborizing within the muscle and reinnervating the muscle fibers. Specifically, ChAT immunoreactive axons of the UN reinnervated the neuromuscular junctions (NMJs) (Fig. 4A, B). Furthermore, this thick section approach corroborated the transmuscular sensory reinnervation as it was possible to observe nerve fibers exiting the muscle and extending into the overlying graft at the musculocutaneous junction (Fig. 5A, B). Nerve fibers in the dermal graft lacked ChAT immunoreactivity, thus confirming their afferent nature. ChAT-positive fibers did not exit the muscle, stopped at the musculocutaneous junction, and then proceeded to run parallel to it (Fig. 6A). Lastly, we found morphological evidence confirming muscle spindle reinnervation seen in the frozen sections. Muscle spindles were identified by their thin intrafusal fiber compared to much thicker extrafusal fibers outside the spindle. A web of nerve fibers surrounding the intrafusal fiber suggested successful reinnervation (Fig. 6B).

## Quantification of sensory reinnervation

Using a sequential retrograde labeling procedure, we were able to determine the degree of sensory reinnervation in the dermal graft at the level of the dorsal root ganglia (DRG) ($n = 9$). In the first step, the UN was labeled with fast blue (FB), a retrograde tracer, before being transferred to the LH. This way, the entire population of pseudounipolar neurons corresponding to the UN's afferent axons were labeled. After a 12-week follow-up period, the second retrograde tracer, Red Retrobeads™ (RB), was applied. The tracer was injected intradermally to be taken up by the afferent nerve fibers reinnervating the dermal graft (Fig. 7). After a one-week follow-up, the DRG and spinal cords (SC)

## NF / BTX / Phalloidin

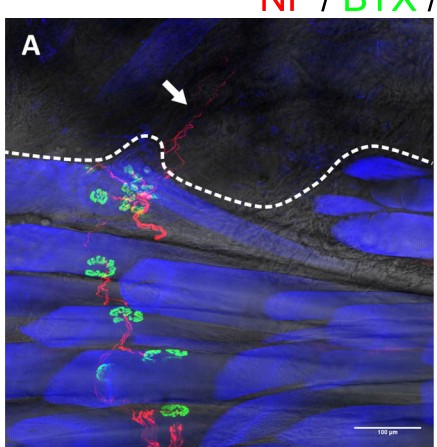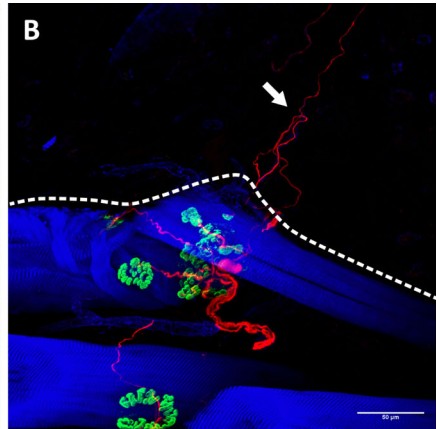

**Fig. 5 | Reinnervation of the dermal graft.** (*n* = 8, biological replicates) The dashed line separates the dermal graft from the underlying muscle. **A** Nerve fiber (NF, red) can be seen trailing between the individual muscle fibers (phalloidin, blue), connecting with the respective NMJs (BTX, green) and eventually extending beyond the muscle into the overlying skin (arrow), suggesting sensory reinnervation. The change in the brightfield image's texture helps with anatomical orientation and confirms the change from muscle to skin. **B** Zoomed-in view of the reinnervating nerve fibers (arrow) without a brightfield image overlay. (NF neurofilament, NMJ neuromuscular junction, BTX bungarotoxin).

were harvested. The samples were cut into serial sections and every section was manually quantified using fluorescence microscopy. The initial tracer visualized the entirety of the UN's neuron population, whereas the second tracer only corresponded to the neurons of the fibers reinnervating the graft, thus making it possible to quantify the degree of reinnervation. All but one C8 DRG were eligible for quantification. That DRG was not quantifiable, as the majority of the tissue was lost during sectioning. Quantification of the remaining DRGs C8 (*n* = 8) and T1 (*n* = 9) revealed 870 ± 175 FB labeled, 132 ± 88 RB labeled, and 113 ± 71 double-labeled (DL) cells and 1118 ± 389 FB labeled, 195 ± 134 (132 (280.5–111.5)) RB labeled, and 150 ± 110 (114 (193–80)) DL cells, respectively. The total cell counts for eight pairs of DRGs were 2019 ± 361 FB labeled, 323 ± 180 (231 (533–184)) RB labeled, and 265 ± 148 DL cells (Fig. 8A–C). Thus, on average, approximately 13% (6–23%) of the UN's neurons in the DRG contributed to the reinnervation of the dermal grafts. Analysis of the SC (*n* = 9) revealed 176 ± 46 FB labeled, 1 ± 1 (0 (0.5–0)) RB labeled and 0 ± 1 (0(0–0)) DL cells (Fig. 8D) (see Supplemental Table S2 for individual cell counts). An unpaired *t*-test demonstrated no significant difference between the total axon count of the UN and the combined FB-labeled cell count of the DRG and SC (*p* = 0.096), providing evidence for the feasibility of using FB in the context of reinnervation processes with longer follow-up periods.

### Deciphering afferent activity of the sensorimotor interface

The functionality of the proposed interface was validated by mechanically stimulating the skin graft while concurrently measuring the afferent electrical activity of the transferred nerve (*n* = 13). Furthermore, the same measuring procedure was performed in a control group which only underwent the nerve transfer procedure without skin grafting (*n* = 10). The stimuli included the determination of touch thresholds using Semmes−Weinstein monofilaments, as well as vibration. After the follow-up period, the muscle (and skin), as well as the transferred nerve were dissected and ENG was performed (Fig. 9A). Afferent nerve activity was measured in 11 of the 13 animals following stimulation with monofilaments. A graded response was observed with increasing monofilament strength, with a mean amplitude of 0.26 ± 0.16 μV, 0.36 ± 0.25 μV, 0.35 ± 0.18 μV, 0.47 ± 0.29 μV, 0.55 ± 0.28 μV, and 0.71 ± 0.52 μV, at 2 g, 4 g, 6 g, 8 g, 10 g, and 15 g, respectively (Fig. 9B and Supplemental Table S3 for individual amplitudes). Furthermore, 11 of the 13 animals showed a clear electroneurographic response to superficial vibration of the skin graft with a

mean amplitude of 1.53 ± 0.85 μV, while no unambiguously distinguishable ENG signals were seen in the other two rats. When applying vibration with a 1-mm indentation, a clearly detectable electrical response was seen in all 13 animals with a mean amplitude of 2.89 ± 1.33 μV. Similarly, electrical signals were recorded in response to undulating vibration in all 13 animals with a mean amplitude of 1.72 ± 0.86 μV (Fig. 9C and Supplemental Table S4). Graded responses to stimulation with monofilaments were found in all animals in the control group (*n* = 10), with mean amplitudes of 0.28 ± 0.11 μV, 0.48 ± 0.28 μV, 0.39 ± 0.16 μV, 0.67 ± 0.36 μV, 0.83 ± 0.57 μV, and 1.16 ± 0.71 μV, at 2 g, 4 g, 6 g, 8 g, 10 g, and 15 g, respectively (Supplemental Table S3). Following a vibration stimulus applied superficially, with a 1-mm indentation or in an undulating fashion, responses were seen in all animals with mean amplitudes of 1.81 ± 1.19 μV, 5.79 ± 3.56 μV, and 4.56 ± 4.07 μV, respectively (Supplemental Table S4). Due to the unexpectedly robust signals measured upon tactile stimulation of the muscle belly in the control group, bouts of five consecutive pulls, as well as constant tension were applied to the distal biceps tendon (Fig. 10A). This served to determine whether the signals were of proprioceptive nature. ENG demonstrated clearly identifiable signal bursts in response to pulling the tendon in all ten control animals with a mean amplitude of 2.31 ± 1.60 μV. Continuous tension applied to the tendon showed an afferent nerve response consisting of an initial peak followed by a plateau for the remainder of the stimulus in all 10 cases as well with a mean amplitude of 2.15 ± 1.82 μV (Fig. 10B). Following these nerve recordings, immunofluorescence staining of thick muscle-skin sections revealed densely reinnervated muscle spindles, as seen in the experimental group with skin grafting. The addition of synaptophysin indicated the presence of nerve terminals in the axon web surrounding the intrafusal muscle fiber (Fig. 6C, D), thus corroborating the ENG findings and the proprioceptive nature of the recorded signals. In order to rule out that the measured ENG amplitudes were the result of movement artifacts of the nerve induced by the tactile stimulation of the interface, the electroneurographic responses to perineural vibration were measured. This stimulus led to a response with a mean amplitude of 0.32 ± 0.20 μV (0.26 (0.44–0.19) μV) (Supplemental Table S4). A Mann−Whitney *U*-test demonstrated a statistically significant difference between the amplitudes measured for superficial and perineural vibration in the control group (*p* < 0.001). This suggests that the recorded signals to different mechanical stimuli indeed originate from the biological interface and are not the result of movement artifacts.

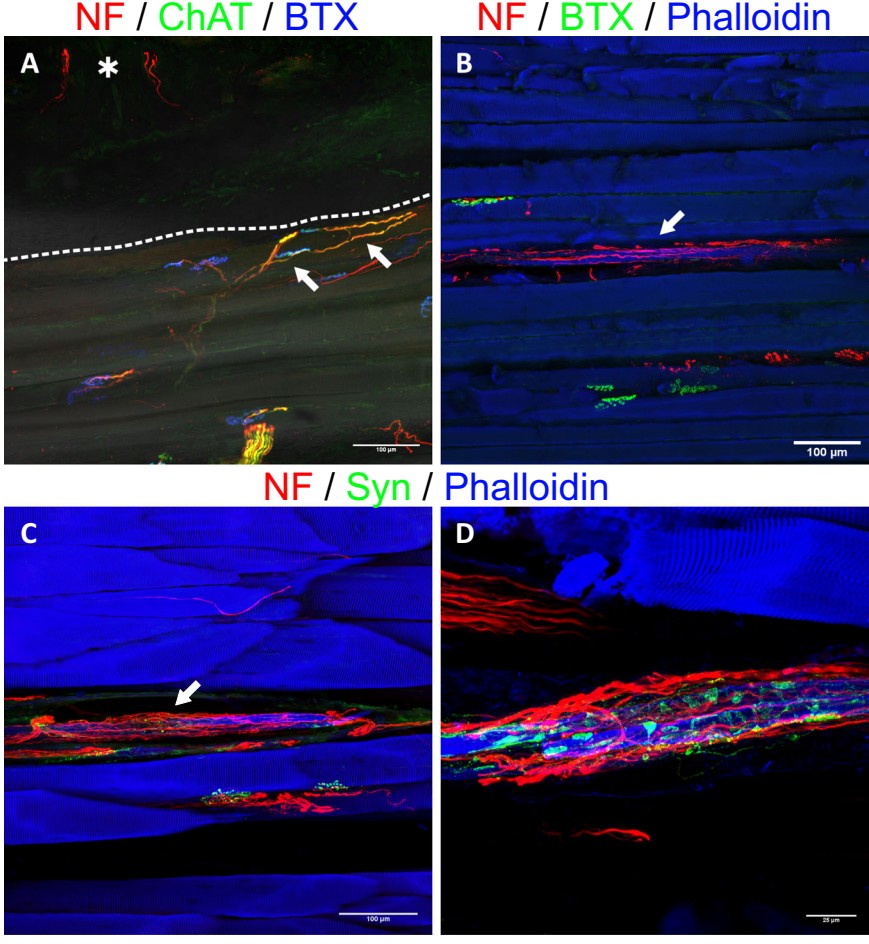

**Fig. 6 | Immunofluorescence staining of 400 µm thick sections from the experimental and control groups.** ($n$ = 8 for all stainings, biological replicates) **A** Anti-NF/ChAT staining (red/green) visualizes efferent axons (yellow, overlay of the two) connecting with NMJs (BTX, blue). The dashed line separates the muscle and skin. This image demonstrates that efferent axons (arrows) extend towards the dermal graft, but do not enter it. ChAT-negative, and thus afferent, fibers can be seen within the graft (*), suggesting specific afferent reinnervation. **B** Morphological evidence of a reinnervated muscle spindle in the experimental group (arrow). The intrafusal muscle fiber (blue) appears slimmer than the other adjacent muscle fibers and is surrounded by a web of nerve fibers (red). **C** Reinnervated muscle spindles were also seen in the control group (arrow). **D** A close-up view reveals numerous synaptophysin-positive (green) structures, suggesting nerve terminals established by the dense web of axons. (NF neurofilament, ChAT choline acetyltransferase, NMJ neuromuscular junction, BTX bungarotoxin, Syn synaptophysin).

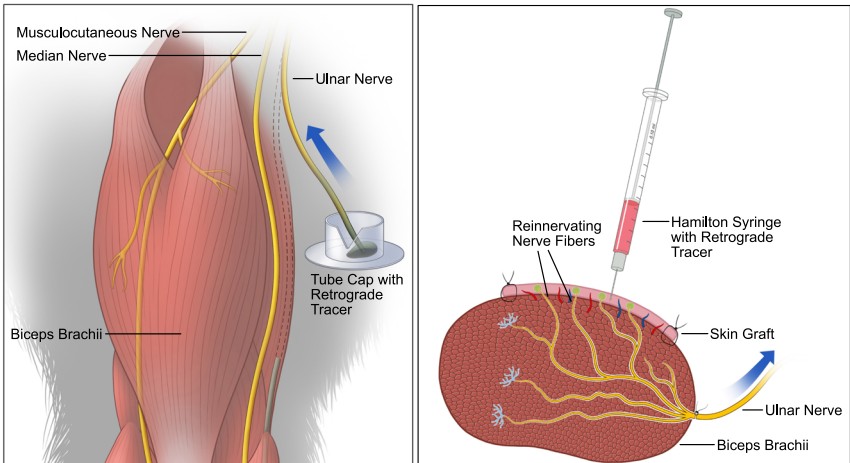

**Fig. 7 | Schematic illustrations of retrograde tracer application.** (Left) The transected UN's distal end is immersed in a dye for 1 h to label its entire neuron population in the spinal cord and DRG (sealing of the v-shaped incision with medical vaseline not shown). (Right) Intradermal injection of the second tracer after the 12-week follow-up period. Reinnervating nerve fibers in the dermal graft take up the injected dye and transport it to the DRG.

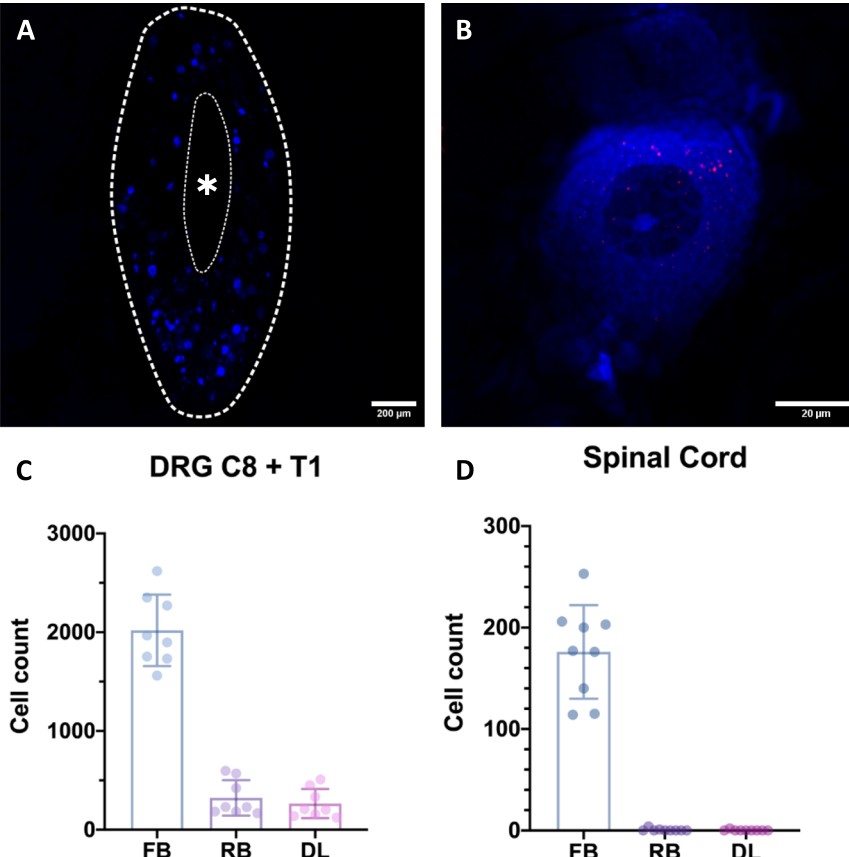

**Fig. 8 | Analysis of the DRGs (*n* = 8, biological replicates) and SCs (*n* = 9, biological replicates).** **A** Overview of an entire DRG cross section (thick dashed line). The individual pseudounipolar cell bodies labeled with FB, as well as their nuclei, are discernible. The asterisk (*) and thin dashed line mark the fibers of the dorsal root. **B** A single-cell body. The nucleus, as well as the nucleolus, are clearly visible. The soma appears blue due to labeling with FB. Individual RBs can be seen in red, indicating tracer uptake via fibers reinnervating the graft. **C**, **D** Overview of the DRG and SC cell counts for each retrograde tracer (FB, RB) and the resulting DL (presented as mean and standard deviation). (DRG dorsal root ganglion, SC spinal cord, FB fast blue, RB Red Retrobeads™, DL double-labeled).

## Discussion

Restoration of sensation in myoelectric prostheses has been attempted using sensory substitution, modality-matched feedback, or peripheral nerve stimulation[15]. It has been demonstrated that all methods can provide a measure of sensation that can support prosthetic control[16], however, many approaches may not be perceived intuitively, thus requiring a re-learning and training process by the patient[15]. Only peripheral nerve stimulation has been shown to be capable of providing afferent stimuli that allow rough topographical allocation to areas of the missing limb (e.g., pressure on the fingertip)[19]. Signal modulation may help transform instances of paresthesia to sensations and even change their intensity or modality[19], but recent data questions the efficacy of modulating the stimulation pattern in this regard[32]. Despite the promising results, the perceivable discrepancy between physiological and artificial sensory feedback (i.e., feeling paresthesia instead of natural sensations) elicited with neural stimulation is due to the inability to recreate the intricate and complex spatiotemporal information that is encoded in afferent signals upon touch or movement[33]. Nevertheless, it should be noted that gross activation of fiber populations resembling the physiological activation pattern provides useful sensory information to the patient and may even be perceived as close to natural due to a certain room for error by the higher-order processing instances in the central nervous system[16,19,33]. The observation of sensory reinnervation following TMR made it possible to interface with the afferent nerve fibers of the missing limb in a somatotopic and potentially more biomimetic way using native cutaneous receptors as mediators. Indeed, cases reported in the literature provide compelling evidence that TSR is a feasible

method to create an intuitive interface for afferent communication[23]. Using reinnervated skin with the appropriate tactile end organs (i.e., mechanoreceptors) as translators to convey afferent information is analogous to the use of reinnervated muscles as biological amplifiers for prosthetic control rather than direct neural interfacing.

Recent work illustrates the increasing interest in biological sensory interfaces and in conveying sensory feedback directly via receptors of the skin rather than electrically stimulating nerves. For example, the concept of RPNIs used for prosthetic control[6] has been modified by either augmenting the construct (composite RPNI = C-RPNI)[34] or replacing the muscle entirely with a dermal skin graft (dermal sensory RPNI = DS-RPNI)[35]. Conceptually, the C-RPNI resembles our proposed interface as it creates a muscle-skin unit for bi-directional communication. Experimental data on C-RPNI demonstrated both efferent and afferent nerve activity, however, the sensory signals were only tested in response to electrical and not mechanical stimulation. In DS-RPNIs, sensory nerves reinnervate a dermal skin graft which then serves as a biological interface, and published results demonstrate afferent responses to both electrical and mechanical stimuli. Another recent experimental approach is the cutaneous mechanoneural interface (CMI)[36] in which a muscle actuator is cuffed around a pedicled sensate skin flap. Electrically stimulating the motor nerve innervating the muscle cuff with implantable components provides the mechanical stimulus acting on the flap and experimental data demonstrated reproducible, graded afferent responses to different stimuli. While the use of pedicled skin flaps may be difficult in an amputation stump due to the limited amount of available tissue, the authors proposed skin grafting onto transected cutaneous nerves,

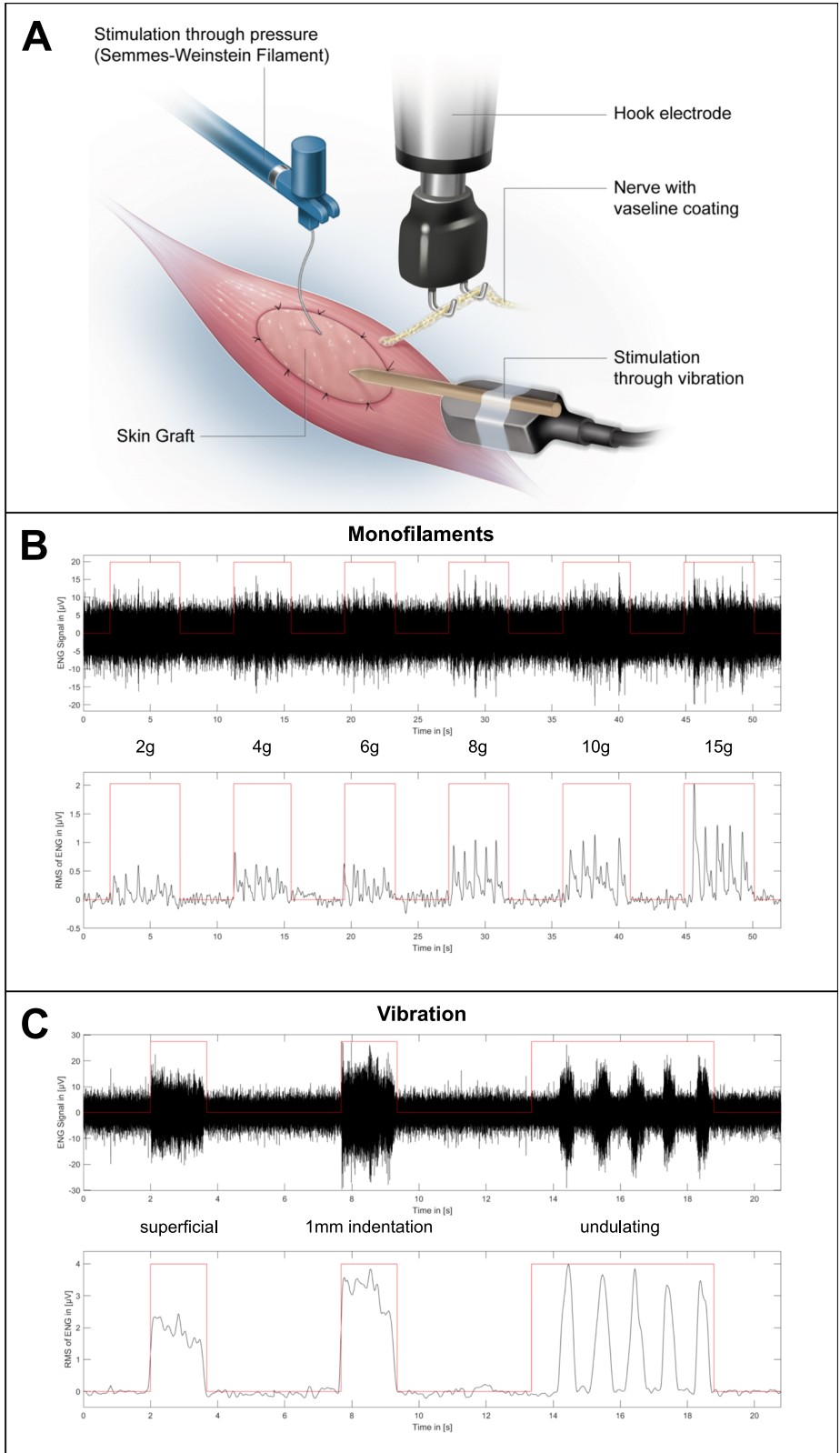

**Fig. 9 | Electroneurographic recordings (filtered ENG signal and its root mean square) after touch and vibration stimulus in the experimental group.** ($n = 13$, biological replicates) **A** The transferred nerve is placed on the hook electrode while mechanical stimuli are applied to the reinnervated skin graft. **B** Stimulating the graft with different monofilaments five consecutive times each led to a reproducible and graded afferent response. **C** Superficial and deep vibration resulted in a robust afferent response. Undulating vibration confirmed the signal's reproducibility.

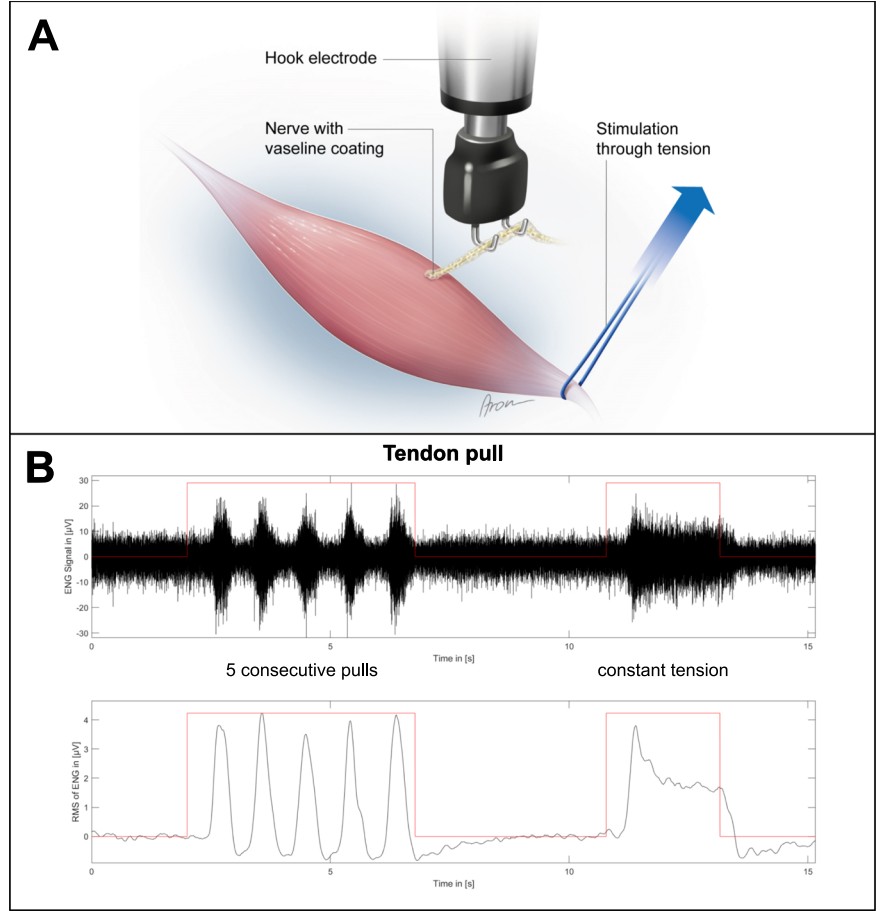

**Fig. 10 | Electroneurographic recordings (filtered ENG signal and its root mean square) after manipulating the distal biceps tendon in the control group.** ($n = 10$, biological replicates) **A** A vessel loop is slung around the distal biceps tendon and pulled while recording the transferred nerve. **B** Consecutively pulling the tendon led to reproducible signal spikes indicating proprioceptive reinnervation. Constant tension applied to the tendon resulted in an initial signal spike followed by a plateau with reduced amplitude. This signal form suggests muscle spindle reinnervation by type Ia fibers.

which conceptually resembles DS-RPNIs. The method proposed in this study directs sensory nerve fibers of a mixed nerve to a dermal graft directly adhering to a muscle targeted for motor reinnervation enabling the translation of mechanical stimuli into afferent nerve activity, thus creating a bi-directional biological interface. While the DS-RPNI and CMI have also been shown to provide sensory responses to touch and vibration, they are not capable of producing proprioceptive signals. This difference may be very relevant as recent evidence demonstrates that fusing touch and proprioception significantly improve prosthetic functionality[25]. Furthermore, they only function as uni-directional interfaces thus requiring the establishment of an additional efferent interface. Next to functional differences, both the C-RPNI and CMI increase surgical complexity as they additionally require muscle grafting and, in the case of the CMI, implantable electronics for stimulation of the muscle actuator. In conclusion, the experimental data regarding the C-RPNI, DS-RPNI, CMI, and our approach is promising, and they may prove to be useful tools in shaping future neuromuscular landscapes for bionic limbs. It seems plausible that the decision on which approach to use may depend on the level of amputation and the individual tissue availability in a given stump. Compared to the currently established TSR method, the placement of the skin graft on top of the muscle creates an insulated motor-skin unit with no other competing native skin afferents, which in turn may help improve the magnitude, precision, and reliability of reinnervation[23]. Furthermore, unlike recent iterations of TSR, this approach is not restricted to superficial muscles directly beneath the skin but can rather be achieved on several muscles within the stump,

opening the possibility of creating a greater array of individual bidirectional interfaces.

The morphological evidence provided in this study using thin sections demonstrated full engraftment of the transplanted dermal graft. It revealed nerve fibers in the skin and muscle indicating both motor and transmuscular sensory reinnervation. Furthermore, it was possible to visualize the Meissner corpuscle and Merkel disc reinnervation, as well as the re-establishment of a dermal plexus. Confocal imaging of stained thick sections revealed the morphology of axonal regeneration, with efferent axons reinnervating NMJs, supporting previous findings of successful motor reinnervation[29]. Afferent axons were again observed extending from the muscle into the overlying skin further supporting the process of transmuscular sensory reinnervation. Lastly, we found compelling morphological evidence for proprioceptive reinnervation. The muscle spindles visualized in thin sections demonstrated an abundance of large, myelinated fibers, while thick section images showed them to be surrounded by a dense web of nerve fibers with nerve terminals suggesting functional sensory reinnervation of these proprioceptors. Interestingly, proprioception is an avenue of sensory feedback that has received comparably little attention in the context of bionic reconstruction of the upper limb despite potentially offering great functional benefits for prosthetic control[25].

Analysis of the DRG following sequential retrograde labeling indicated specific cutaneous reinnervation by the transferred UN. Furthermore, to our knowledge, this study provides the first quantification of targeted reinnervation at the level of the DRG. There was a

noticeable variability regarding the degree of reinnervation, ranging from approximately 6–23%. This is most likely caused by a random and currently uncontrollable reinnervation process, an observation also made in TSR patients with largely varying cutaneous projections of the missing limb[22]. Analysis of the SC revealed close to no DL motor neurons, indicating virtually no retrograde tracer uptake via motor fibers thus supporting the dermal grafts purely sensory reinnervation, as well as the specificity of this labeling procedure. Additionally, the axon quantification confirmed the feasibility of FB for investigating reinnervation as there was no statistically significant difference between the number of axons in the periphery and neurons in the DRG and SC. This suggests no noticeable drop in FB-labeled cells after 12 weeks supporting previous findings[37]. The axon counts may also serve as reference values for the donor and recipient nerves to help guide researchers in developing novel nerve transfer models in future studies. Furthermore, the distribution of efferent and afferent nerve fibers supports previous work investigating the fiber composition of nerves in the upper extremity of human cadavers[38]. Another important finding in regard to ethical considerations when planning animal trials is that harvesting autologous dermal grafts for sensory reinnervation is as feasible as using donor animals[34], as demonstrated by the low donor site morbidity and lack of locomotive deficits in our study both postoperatively and in the following 12 weeks.

The electroneurographic evidence presented in this study corroborated the morphological findings of sensory reinnervation. Tactile stimulation of the biological interface with monofilaments led to a graded response indicating increasing recruitment of reinnervated receptors corresponding to greater mechanical force. Furthermore, vibration stimuli produced afferent nerve signals with larger amplitudes following deeper (i.e., 1 mm indentation) application, also suggesting an increased receptor recruitment with greater vibrational force. Interestingly, robust and reproducible electroneurographic activity was also seen in the control group following mechanical stimulation with touch and vibration. These unexpected afferent signals were thought to be of proprioceptive origin, as there was no reinnervated skin graft present. Subsequently, manipulation of the biceps' distal tendon revealed clearly identifiable electrical responses to brief or constant pulling of the tendon. Indeed, when applying constant force, the signal displayed an initial spike followed by a plateau with lower amplitude and concluded with a small dip below baseline after release. This corresponds to dynamic and static signal components most likely originating from type Ia fibers[39] of reinnervated muscle spindles. The immunofluorescence stainings using anti-MBP antibodies further supported the presence of this fiber type as it visualized large, myelinated nerve fibers both within muscle spindle capsules, as well as in direct proximity accompanying them. In conjunction with the reestablished nerve terminals on the intrafusal fibers of muscle spindles, this provides a strong case for the reproducible and robust functional reinnervation of proprioceptive organs, thus expanding the possibilities of afferent feedback in bionic prostheses. Overall, the biological sensory interface demonstrated reproducible afferent activity in response to different mechanical stimuli. While our morphological evidence demonstrated sensory reinnervation of the skin graft both intradermally and at the level of the DRGs, it is difficult to separate the signal contributions resulting from reinnervated cutaneous and the proprioceptive afferents as the graft is attached to the muscle, and thus mechanically coupled. In fact, identifying the individual components of electroneurographic signals obtained with electrodes recording compound action potentials from peripheral nerves in vivo remains challenging not least because the amplitudes are in the single or low double digit μV range resulting in inherently low signal-to-noise ratios[40]. Furthermore, closer examination of the reinnervating nerve fibers' morphology revealed that many fibers found in the dermal graft were unmyelinated despite mechanoreceptors found in native glabrous skin being innervated primarily by myelinated fibers[41]. It is possible that the musculocutaneous junction poses a mechanical barrier for reinnervating myelinated fibers. In any case, recording unmyelinated fibers with extraneural methods remains challenging and may only be feasible with more invasive methods[42]. This property may have also added to the difficulty in separating the different afferent signal components. A solution may be the application of artificial intelligence classifiers to distinguish between cutaneous and proprioceptive signals[40]. Thus, future studies specifically investigating the signal contribution of different sensory modalities in the context of targeted reinnervation are needed.

The presented findings are promising for establishing biological sensorimotor interfaces and experimental data suggests that the restoration of tactile sensation, and thus the utilization of a closed-loop control system, may improve prosthetic control, however, further research is needed to measure and confirm its contribution, as well as to elucidate the role of feedforward systems more accurately[43,44]. Regardless, sensory feedback remains a concern for many patients[13], and TMR and TSR have been demonstrated to help alleviate phantom limb and neuroma pain, as well as improve the embodiment of the prosthesis, respectively[45,46]. Another important aspect of sensory restoration in bionic limbs is achieving proprioceptive feedback. Current approaches range from non-invasive modalities[47,48] to invasive peripheral nerve stimulation[49,50]. Furthermore, the agonist-antagonist myoneural interface[51], as well as vibration applied to tendons[52] or to muscles following targeted reinnervation[53] have been shown to elicit proprioception, thus opening an avenue to directly interact with physiological receptors. Despite these efforts, proprioception remains an underrepresented modality compared to exteroception in the body of research pertaining to sensory feedback in the upper extremities. Recent data suggests that the integration of both tactile feedback and proprioception may significantly increase dexterity, approaching the performance of able-bodied subjects[25]. In the context of targeted reinnervation, it is unclear to which degree and by which afferent fibers' proprioceptive end organs are reinnervated. The evidence presented in this study suggests muscle spindle reinnervation is most likely by type Ia afferents, however, more research is needed to investigate the exact mechanism, as well as the reinnervation of Golgi tendon organs[54] or even cutaneous receptors in reinnervated skin (grafts) near (or from) joints[22,55,56]. Lastly, research is needed on how to interact with the array of new biological sensory interfaces to enable translation into clinical reality. As traditional TSR can be combined with non-invasive feedback systems[24,25], the presented biological interface may also work with external devices or even implantable components stimulating the skin graft. A possible solution may be an implantable magnetic mesh actuated by an external electromagnetic system, similar to a recently proposed concept for transmitting sensory information[57]. As pulling a tendon will not be a feasible way to elicit proprioceptive feedback in a stump and vibration can induce proprioceptive activity as demonstrated in this study and other recent work[25], the implantation of magnets into the skin graft, muscle belly and potentially even the tendon may be a solution to interact with the pathways for both cutaneous and proprioceptive feedback.

In conclusion, to our knowledge, this is the first established animal model with a detailed multilevel morphological, as well as electrophysiological investigation of a biological, sensorimotor interface. We proposed the possibility of epimysially placing a glabrous dermal skin graft in conjunction with targeted reinnervation and demonstrated the skin's engraftment, as well as dermal, mechanoreceptor, and muscle spindle reinnervation. Furthermore, we were able to quantify the degree of reinnervation by employing a sequential retrograde labeling procedure and confirmed the interface's functionality with ENG. The morphological and electrophysiological data also granted insight into the basic neurophysiology underlying the process of targeted reinnervation for the first time. This approach bears great potential for creating biological interfaces in future neuroprosthetic applications,

as it grants access to the information flow from and to the extremity by re-establishing both the peripheral efferent and afferent pathways directly related to sensorimotor control of the hand. The combination of this biological tactile and proprioceptive sensory interface with multiple nerve transfers[58] and implantable multichannel electromyography electrodes[5] may enable the creation of a "bio-hub", thus greatly improving the man-machine interface in bionic reconstruction and ultimately helping patients regain their lost extremity function.

## Methods

### Study design

The aim of this study was to surgically create a bi-directional, biological interface in order to close the communication loop in neuroprosthetic applications. By utilizing selective nerve transfer and skin transplantation, we hypothesized that a single muscle-skin unit could provide reinnervated end organs for efferent and afferent communication. The project was approved by the institutional review board and the Austrian Federal Ministry of Education, Science and Research (BMBWF 2020-0.171.173).

### Animals

Thirty-two male Sprague-Dawley rats (Charles River Laboratories, Germany) aged 8–10 weeks were used in this study and divided into three groups:

1. Surgical procedure using skin graft for tissue examination and ENG ($n = 13$).
2. Surgical procedure using skin graft for sequential retrograde labeling ($n = 9$).
3. Surgical procedure without skin graft serving as a control group ($n = 10$).

Rats were held under standardized housing conditions in cages with a 12-h day and night cycle, acclimated for two weeks before any interventions, and had access to water and food ad libitum. All animals received a mixture of water, glucose, and piritramide (30 mg piritramide, 30 ml 10% glucose, and 250 ml water) as analgesia for the first postoperative days. The standard follow-up period was 12 weeks and 12 + 1 weeks for the sequential retrograde labeling group. Housing and all procedures were done according to the guidelines of the Federation of European Laboratory Animal Science Associations[59].

### Surgical procedure

Anesthesia was induced by administering ketamine (100 mg/kg body weight) and xylazine (5 mg/kg body weight) intraperitoneally and then maintained with inhalation of 1.5% isoflurane via an endotracheal tube. Analgesia was achieved by subcutaneous injection of piritramide (0.3 mg/kg body weight). The rats were positioned in a supine position with both arms fixed at 90° abduction. An approximately 2.5 cm long incision was made from the acromion to the medial epicondyle (Supplemental Fig. S4A). Subsequently, the LH was microscopically dissected, and its fascia was removed (Supplemental Fig. S4B). The biceps' two heads were gently separated to expose the LHMB (Supplemental Fig. S4C), which was subsequently dissected proximal and transected shortly before its origin from the musculocutaneous nerve. Next, the UN was located proximal to the cubital tunnel just medial of the medial intermuscular septum. The UN's position beneath the triceps' medial head can be identified by its accompanying vein (Supplemental Fig. S4D). The UN was exposed through a small window in the muscle and dissected both proximally and distally before transecting it just proximal to the cubital tunnel. Afterwards, it was identified proximally in the medial bicipital groove (Supplemental Fig. S4E). After dissection and distal mobilization, the UN was easily pulled out underneath the triceps brachii and then transferred directly to the motor entry point of the LH with one 10-0 nylon (Ethicon, USA) single interrupted suture (Supplemental Fig. S4F). In the control group, the

skin was closed by 6-0 absorbable dermal and simple interrupted sutures, thus finishing the procedure at this point. In the groups using a skin graft, the operating site was covered with a wet swab followed by harvesting of the glabrous dermal skin graft from the ipsilateral hindlimb. The graft was marked between the walking pads (Supplemental Fig. S5A). The area was then subcutaneously injected with saline to increase the skin's tension and was carefully deepithelized using a standard 15-blade. The marked borders were incised, and the proximal end was grasped with a tweezer to separate the graft from the plantar fascia. The defect was primarily closed with 6-0 absorbable inverted simple interrupted sutures (Supplemental Fig. S5B, C), due to the thin plantar skin and to minimize irritation caused by knots while walking. Subsequently, the graft was defatted, placed on the LH's surface with the hypodermis facing the muscle, and fixed epimysially using 10-0 nylon simple interrupted sutures (Fig. 1). Lastly, the skin was closed as described before. After the follow-up period, rats were deeply anesthetized, and native, contralateral UN and LHMB samples were harvested from the contralateral extremity. This was followed by intracardial perfusion with 300 ml 0.9% sodium chloride solution and 400 ml 4% paraformaldehyde (PFA) and harvesting of the muscle-skin samples.

### Sequential retrograde labeling

A sequential retrograde labeling procedure was employed to quantify the degree of sensory reinnervation in the skin graft at the level of the DRG. The steps of the surgical procedure were performed as described before, however, the UN was subjected to retrograde labeling with FB (Polysciences, USA) before being transferred to the LH. For this purpose, a v-shaped incision was cut into the cap of an Eppendorf tube (Safe-Lock tube, Eppendorf, Germany) which then served as the reservoir for the tracer. By creating this cut-out, the nerve can lay flatly in the cap without kinking. The incision was covered with medical vaseline (Fagron, Germany) to protect the nerve from being damaged by any rough edges. Five microlitres of the retrograde tracer were injected into the cap using a pipette and the cap was securely placed in a small pocket medial to the medial bicipital groove. The proximal nerve stump was then gently placed into the dye and the v-shaped incision was sealed off with vaseline to prevent tracer leakage. The nerve was labeled for 1 h (Fig. 7) while the operating site was covered to minimize light exposure and the glabrous dermal skin graft was harvested during this time. While the nerve was still being stained, the cap with the nerve and tracer was covered with several swabs to allow fixation of the graft to the LH's epimysium. The nerve transfer was then performed as described. After the 12-week period, the rats were anesthetized again, and the skin graft was dissected. A second retrograde tracer was injected intradermally in approximately 2 mm intervals using a Hamilton syringe with a 30 g needle (Fig. 7). For this purpose, RB (Lumafluor Inc., United States) were chosen as they offer a good contrast to the initial tracer and only spread within an approximately 1 mm radius from the injection site[60], thus preventing the tracer to seep into the muscle. Furthermore, close care must be taken to not inject the tracer into the muscle, as this may possibly stain pseudounipolar neurons corresponding to proprioceptive afferent fibers. This way, only nerve fibers reinnervating the graft retrogradely transport the tracer, therefore labeling the respective pseudounipolar neurons and allowing analysis of DL cells. Following the injection, the wound was closed, and the rats recovered for another week.

### DRG and spinal cord

Following perfusion, a median incision was placed along the spinous processes, and the spine was freed from the surrounding muscle and connective tissue. The spinous processes were removed with a bone rongeur to expose the SC. Subsequently, the laminae, as well as the articular and transverse processes on the right side were carefully removed to expose the DRG. The DRG C8 and T1, as well as the SC

segments C6 to T2 (each $n = 9$) were harvested and immediately stored in 4% PFA tubes wrapped in aluminum foil to prevent any further light exposure. They remained immersed in PFA for up to an additional 8 h before being rinsed with phosphate-buffered saline (PBS) for 24 h. This was followed by dehydration in three glucose mixtures of increasing concentrations (10%, 25%, and 40% in PBS) for 24 h each. The samples were embedded in an optimal cutting temperature compound and stored at −80 °C before sectioning. Frozen samples were cut into 20 μm serial sections with a cryostat (CM3050 S, Leica Microsystems, Germany) and picked up on microscope slides. All slides were then stored at −80 °C until further analysis.

## Histology

Fixation and rinsing of muscle skin (experimental group), muscle (control group), and nerve (native) samples with PFA and PBS were performed as described before. The muscle-skin samples were cut into approximately two halves and one thin slice was taken from the middle. One half was embedded in paraffin for H&E staining. For this purpose, samples were placed in a tissue cassette, dehydrated (35 min in 100% ethanol and 90 min in isopropanol and paraffin each) using a microwave tissue processor (KOS, Milestone, Italy), and then embedded in paraffin. Afterwards, the samples were cut into 4 μm sections and bound to microscope slides. The sections were then subjected to H&E staining. In short, the slides were briefly warmed in an oven at 56 °C, then rinsed in HistoSAV (SAV Liquid Production GmbH, Germany) (1 × 20 min, 1 × 10 min), 100% ethanol (2 × 2 min), 96% ethanol (2 × 2 min), 80% ethanol (2 min), 70% ethanol (2 min) and distilled water (5 min). This was followed by immersion in Mayer's hemalum solution (3 min), 0.1% hydrochloric acid (2 s), running tap water (3 min), eosin-Y solution (alcoholic) (3 min), and another round of running tap water (30 s). Lastly, the slides were rinsed with 96%, 100% ethanol, and HistoSAV (2 × 10 s each) before applying a mounting medium and covering them with coverslips.

## Immunohistochemistry

The muscle-skin slices and native UN and LHMB samples were dehydrated as described before. They were embedded, sectioned, and stored like the DRG and SC. Frozen muscle-skin ($n = 13$) and nerve ($n = 6$) sections were then stained within a few weeks to prevent tissue degradation and visualize reinnervating nerve fibers in the skin graft and muscle, mechanoreceptors, and muscle spindles, as well as to quantify the amount of afferent and efferent (cholinergic) nerve fibers, like previous work[30], respectively. The slides were thawed at room temperature for approximately 30 min before being immersed in PBS three times for 5 min each. The individual sections on each slide were then encircled by a hydrophobic barrier and blocked with 10% rabbit serum (Dako, Agilent Technologies, USA) (in PBS-Triton (PBST)) for 1 h at room temperature. Afterward, the serum was quickly shaken off and muscle-skin sections were incubated with a chicken anti-NF primary antibody (Merck Millipore, USA) solution (1:2500 in PBST) for 48 h at 4 °C, while nerve sections were additionally incubated with a goat anti-ChAT primary antibody (Merck Millipore, USA) solution (1:100) to visualize nerve fibers. The slides were then again rinsed in PBS three times for 5 min each. Muscle-skin sections were then incubated with a rhodamine rabbit anti-chicken secondary antibody (Invitrogen, Thermo Fisher Scientific, USA) (1:200 in PBST) solution for 2 h at room temperature, while nerve sections were additionally incubated with an Alexa Fluor 488 rabbit anti-goat secondary antibody. Subsequently, the slides were rinsed with PBS, PBST, and twice more in PBS for 5 min each. Lastly, a fluorescence mounting medium (Dako, Agilent Technologies, USA) was applied to the slides which were then covered with coverslips. Mechanoreceptors and muscle spindles were stained using 10% goat serum (Dako, Agilent Technologies, USA) in combination with a Dako Omnis rabbit anti-S100 primary antibody (Dako, Agilent Technologies, USA) solution (1:1 in PBST) or a mouse anti-MBP primary

antibody (Santa Cruz Biotechnology, USA) solution (1:500 in PBST), respectively, in addition to the aforementioned anti-NF primary antibody. Alexa Fluor 488 goat anti-rabbit, 488 goat anti-mouse, and 568 goat anti-chicken secondary antibody solutions (1:400) were used while keeping all other steps of the protocol the same. Immunofluorescence staining of the other muscle-skin sample halves, as well as reinnervated muscles from the control group, were performed similarly to a protocol previously described[31]. In short, samples were stored in PBS with 0.05% sodium azide for one to two weeks and then cryoprotected in three sucrose mixtures of increasing concentrations (10%, 25%, 40%), embedded in a cryomatrix (Thermo Fisher Scientific, USA) and stored at −80 °C. 400 μm thick sections were cut parallel to the longitudinal axis of the muscle-skin sample within a few weeks. Floating sections were then stained with 3 different triple-labeling antibody combinations consisting of (1) anti-NF, BTX, phalloidin; (2) anti-NF, anti-ChAT, BTX, and (3) anti-NF, anti-Syn, phalloidin. Anti-NF is a general neural marker, anti-ChAT is a marker for cholinergic axons and anti-Syn is a marker for nerve terminals. BTX and phalloidin are toxins that visualize NMJs and muscle fibers, respectively. Before staining, samples were incubated in PBST for 24 h and then blocked with 10% normal goat serum (staining combinations 1 and 3) or rabbit serum (combination 2). This was followed by incubation with the primary antibodies for two days. This was followed by rinsing with PBST, incubation with Alexa Fluor conjugated secondary antibodies (1:500 in PBST), phalloidin (1:200) or BTX (1:500) for 4 h, another rinsing with PBST, and lastly mounting in fluorescence mounting medium.

## Imaging and tissue analysis

Quantification of the labeled pseudounipolar and motor neurons in the DRG and SC was performed manually using a TissueFAXs slide scanner (TissueGnostics, Austria) at 20–40× magnification. All sections were analyzed consecutively using the DAPI filter to detect cells labeled with FB, while the Texas Red filter was used for RB-labeled cells. In case of missing sections, the average of the previous and following cell counts was used. Cells were counted when tracer uptake and nucleolus were visible. Cells displaying uptake of both tracers were considered DL. As the RB tracer consists of individually visible spheres, a cell was considered labeled when five or more beads were visible within the cell body, similar to previous work[61]. The Abercrombie formula[62] was used to correct for cells counted twice when split between two sections. For this purpose, the average nucleolus diameters for pseudounipolar and motor neurons were determined. A fluorescence image of each sample's section with the highest total FB cell count was acquired with the aforementioned slide scanner, five randomly chosen nucleoli were measured using Fiji[63], and their mean value was calculated (Supplemental Table S5). DRG and SC images were acquired with an LSM700 confocal microscope (Zeiss, Germany) using the 405 nm and 555 nm lasers to visualize the FB and RB tracers, respectively. Image acquisition of muscle-skin cross sections was also performed with an LSM700 using the 488 nm and 555 nm lasers for MBP/S100 or NF, respectively. Images of UN and LHMB cross-sections were acquired using a TissueFAXs slide scanner at 20× magnification. NF and ChAT were visualized using the Texas Red and FITC filters, respectively. Muscle-skin images were qualitatively analyzed, while all nerves were manually quantified using Fiji's cell counter function[63]. Images of stained 400 μm thick sections were acquired using an FV3000 confocal laser scanning microscope (Olympus, Germany). The triple-labeled samples were visualized using lasers with 488 nm, 568 nm, and 633 nm wavelengths. Furthermore, brightfield images were also acquired and merged with fluorescence images to enable better anatomical orientation. Images were then qualitatively analyzed.

## Tactile stimulation and ENG

The reinnervated biological interfaces (experimental group $n = 13$, control group $n = 10$) were mechanically stimulated (touch thresholds

and vibration in both groups, proprioception in the control group) while recording afferent nerve activity of the transferred nerve to assess their functionality. First, the operating site was reopened via the previous incision. Both the muscle-skin unit and the transferred nerve were dissected. The pectoralis major was then excised to expose the proximal part of the UN to ensure sufficient length for the placement of the electrodes. Subsequently, the nerve was carefully freed from any connective tissue to minimize interference during ENG. Afferent nerve activity was recorded using custom-made bipolar hook electrodes made of tinned copper alloy with a 5 mm gap between the hooks. The nerve was carefully placed onto the hooks without any tension and covered in medical Vaseline (Fagron GmbH & Co. KG, Germany) to prevent it from drying. An 18 G hypodermic needle was placed subcutaneously close to the recording site and acted as a ground electrode. The electrodes were connected to a dedicated pre-amplifier (Neuro Amp EX Model FE285, ADInstruments Inc., USA) which was hooked up to a data acquisition system (PowerLab 16/35, ADInstruments Inc., USA). Data was recorded at a sample rate of 40 kS/s and subsequently filtered in two stages. First, an analog bandpass filter (300–5000 Hz) was implemented in the pre-amplifier followed by the application of an additional digital bandpass filter (800–3200 Hz) to further improve signal quality. All data was then processed in MatLab (R2010a, The MathWorks Inc., USA) and root mean square values (200 ms moving window) were computed. The resulting maximal ENG amplitudes were then used for qualitative and quantitative analyses. Semmes–Weinstein monofilaments of increasing strength (2 g, 4 g, 6 g, 8 g, 10 g, and 15 g) (Baseline Measurement, Fabrication Enterprises, USA) were used in bouts of five contacts each to apply mechanical pressure to the graft and determine the sensory threshold. Vibrational bursts of 2 s were applied either superficially (i.e., touching without additional pressure) or with a 1 mm indentation using a custom-made vibrating device operating at a frequency of 100 Hz. This was followed by the application of an undulating vibration stimulus (briefly touching the graft five times in a row). Both touch and vibration stimuli were applied to the skin grafts center in the experimental group or the corresponding position on the muscle belly in the control group (Fig. 9). Lastly, a vessel loop was slung around the distal biceps tendon in the control group to pull the tendon thus investigating the proprioceptive response. The stimulation sequence consisted of five short, consecutive pulls followed by a prolonged stretch of approximately 3 s. This way, the dynamic and static components of the proprioceptive response could be distinguished (Fig. 10). Lastly, vibrational stimuli were also applied approximately 1 mm away from the nerve coaptation site at the motor entry point in the control group. This perineural vibration simulated an intense mechanical interference to determine whether the tactile stimuli applied to the skin or muscle were capable of inducing movement artifacts during recording. These signals were then compared to superficial vibration stimuli which yields the lowest amplitudes and would therefore be most susceptible to artifacts. A trigger button was used to mark the application of each stimulus within the recorded data.

### Statistical analysis
Descriptive statistics were calculated for the axons in the harvested nerves, the maximal ENG amplitudes, as well as for the pseudounipolar and motor neurons in the DRG and SCs, respectively. The data was assessed for normal distribution using the Kolmogorov-Smirnov test. It is reported as mean and standard deviation and additionally median and interquartile range (Q3–Q1) in case of non-normality. Depending on the data's distribution, an unpaired $t$-test or Mann–Whitney $U$-test was performed to compare the total axon count in the peripheral UN with the total cell count in the DRG and SCs labeled with FB to validate the tracer's use in long-term follow-up. Furthermore, the amplitudes between superficial and perineural vibration in the control group were compared using the same approach. The level of significance was set at

$\alpha = 0.05$. All statistical analyses were performed in Prism (version 9.0.2; GraphPad Software, USA) and SPSS (version 29.0.0.0; IBM, USA).

### Reporting summary
Further information on research design is available in the Nature Portfolio Reporting Summary linked to this article.

## Data availability
All data supporting the findings of this study are available within the article and its supplementary files. Any additional requests for information can be directed to, and will be fulfilled by, the corresponding authors. Source data are provided with this paper.

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

## Acknowledgements
We would like to thank Anna-Maria Willensdorfer for her invaluable technical support. Furthermore, we want to thank Aron Cserveny (https://www.sciencevisual.at/) for contributing exceptional illustrations to visualize the key concepts of this study.

## Author contributions
C.F., D.F., K.D.B., and O.C.A. developed the concept. C.F., V.T., M.S., M.L., K.D.B., and O.C.A. designed the study and experimental model. C.F., J.O., U.M., V.T., M.S., and M.L. collected the samples and data. C.F., J.O., U.M., R.B., and G.C.R. processed the samples and performed microscopy. C.F., R.B., M.S., G.C.R., G.L., D.F., K.D.B., and O.C.A. analyzed and interpreted the data. C.F., D.F., K.D.B., and O.C.A. drafted the manuscript. All authors revised and approved the final version of the manuscript.

## Funding
This project has received funding from the European Research Council (ERC) under the European Union's Horizon 2020 research and innovation programme (grant agreement no. 810346).

## Competing interests
All authors declare no competing interests relevant to the conception and conduction of this study or the interpretation of its results.
