## [Peer Review File · Nature Communications]

REVIEWER COMMENTS

Reviewer #1 (Remarks to the Author):

This study introduces a fairly novel surgically created biologic interface to transduce sensory signals from a prosthesis that involves performing TMR and placing a glabrous skin graft on the target muscle such that the sensory axons regenerating into the target muscle continue on to reinnervate the skin graft. This is a well executed study that provides proof of concept that a skin graft overlying a reinnervated muscle will also become reinnervated by sensory axons and that mechanical stimuli triggers afferent signal propagation through the donor nerve.

The novelty of this approach is somewhat incremental in that TSR, C-RPNI, and CMI have previously been described. The authors discuss some theoretical benefits of their proposed approach, but they are not specifically tested in the presented dataset. For example, the authors posit that a glabrous skin graft will have a greater density of reinnervated Meissner corpuscles than the native skin overlying a reinnervated muscle, which in turn will presumably provide higher quality sensory feedback, but this hypothesis was in no way tested. Likewise, they do not demonstrate the benefits of glabrous vs non-glabrous (as in CRPNI) skin grafts for this purpose. While the high quality data set they provide does strengthen the evidence supporting the viability of placing a skin graft over the target muscle in TMR, it remains unclear whether their modified approach demonstrates a clear step forward. Inclusion of representative control groups for the previously described approaches would have been helpful in this regard.

Additional Comments:

No evidence was provided to suggest reinnervation of specific sensory organelles (Meissner's corpuscles, etc) within the glabrous skin graft. This weakens the case for using this relatively scarce skin type for as a graft vs non-glabrous skin grafting or the native skin overlying the target muscle.

Direct mechanical stimulation of the target muscle without skin graft (the only control group) demonstrated greater amplitude ENR signals on average than mechanical stimulation of a glabrous skin graft overlying the target muscle. This could be explained by the reinnervated spindles within the target muscle which were nicely demonstrated by the authors. This finding also raises the possibility that the skin graft itself did not contribute to mechano sensitive signal transduction and may have actually provided a shielding effect of the mechanosensitive muscle itself. Again, the authors did not demonstrate reinnervation of mechanosensitive organelles in the skin grafts.

Reviewer #2 (Remarks to the Author):

The manuscript by Festin and colleagues reports on the creation of a novel biological sensorimotor interface for potential bionic reconstruction in a rat model. Overall, the rationale behind the paper is sound, and the results are encouraging. However, there are several limitations which decrease enthusiasm for the paper.

The authors state that their model is novel, however, it is not very different from other models already published (i.e., VDMT, C-RPNI). The authors do acknowledge this, but it limits overall enthusiasm for the novelty of the construct. Specifically, their construct is very similar to VDMT since they take an intact muscle and denervate it, but they leave the vascular pedicle and also leaves the muscle insertion to the bone intact. It is not clear what the construct even is until you read the methods section. It was unclear whether it was an intact muscle with intact neurovascular structures or a muscle graft or some other surgical technique.

The primary concept of the paper is a "bidirectional" interface. However, the results do not discuss anything about the ability of the interface to facilitate motor control. On the flip side, while they discuss electrophysiological sensory results, there is no immunohistochemistry showing reinnervation of sensory end organ targets in the skin. The figures showing "reinnervation of the skin" are not showing that the fibers are coming from the muscle side. If the skin is really reinnervated by the transferred nerve, it would show as fibers coming from the muscle into the skin.

There are no postoperative photos of the construct showing that there is survival of the denervated muscle and the overlying skin graft. If there are really a lot of cysts like they said in the text, from a clinical perspective, wouldn't a secondary surgery be needed to remove all the cysts and wouldn't this cause a lot of pain given c-fiber transection during cyst removal? And another surgery for electrode implantation once this heals?

There is an issue with deciphering afferent activity of the sensorimotor interface. The authors used a construct without skin graft (so muscle only) as their control. But this control construct had robust signals to light touch and vibration. The authors concluded that the response to the monofilament and vibration was proprioceptive in nature which doesn't make sense as muscle spindles are not directly involved with responding to fine touch sensations like those detected by cutaneous mechanoreceptors. How do they know that instead, it isn't a result of direct activation of the nerve? How does this compare to normal intact innervated skin or normal intact innervated muscle? Furthermore, there are no stats provided comparing to the control.

The idea of the construct having proprioceptive signaling abilities is a strong point of the paper. However, pulling on the muscle will inevitably pull on the nerve and thus create a movement artifact. How are the authors certain that the readings they are getting are actually proprioception in nature and not simply movement artifact?

There are some issues with the retrograde labeling results. First, it is very hard to see the RB's in Figure 6. It is unclear as to why the authors didn't use a secondary tracer like FluroRuby, which would have also stained the entire cell body like FB. In the current study, how do the authors know that the RB labeled DR cells are all from nerves reinnervating the dermal graft? The skin graft is very thin and it is definitely possible for the dye to be picked up by the muscle and thus it could be from afferent nerves with muscle origin.

REVIEWER COMMENTS

Reviewer #1 (Remarks to the Author):

This study introduces a fairly novel surgically created biologic interface to transduce sensory signals from a prosthesis that involves performing TMR and placing a glabrous skin graft on the target muscle such that the sensory axons regenerating into the target muscle continue on to reinnervate the skin graft. This is a well executed study that provides proof of concept that a skin graft overlying a reinnervated muscle will also become reinnervated by sensory axons and that mechanical stimuli triggers afferent signal propagation through the donor nerve.

RE: Thank you for the summary and positive feedback.

The novelty of this approach is somewhat incremental in that TSR, C-RPNI, and CMI have previously been described. The authors discuss some theoretical benefits of their proposed approach, but they are not specifically tested in the presented dataset. For example, the authors posit that a glabrous skin graft will have a greater density of reinnervated Meissner corpuscles than the native skin overlying a reinnervated muscle, which in turn will presumably provide higher quality sensory feedback, but this hypothesis was in no way tested.

RE: We agree that our approach builds on TSR, but it certainly has specific characteristics that differentiate it from previous approaches. Regarding Meissner corpuscles, for example, current evidence suggests that these mechanoreceptors are located in glabrous skin (1), meaning that a comparison with the density of the overlying native skin (typically non-glabrous for muscles used in TMR) is not useful. Nonetheless, we have substantially extended our analyses. Our newly provided evidence demonstrates reinnervation of Meissner corpuscles (lines 156-157, figure 2B) and even suggests the re-establishment of a dermal plexus with Merkel discs (lines 156-157, figure 2C, D), which has not been described before in other approaches.

Likewise, they do not demonstrate the benefits of glabrous vs non-glabrous (as in CRPNI) skin grafts for this purpose. While the high quality data set they provide does strengthen the evidence supporting the viability of placing a skin graft over the target muscle in TMR, it remains unclear whether their modified approach demonstrates a clear step forward. Inclusion of representative control groups for the previously described approaches would have been helpful in this regard.

RE: We appreciate the recognition of the quality of our data. It should be noted that C-RPNIs published in the literature (2) also used glabrous skin. We have added analyses to demonstrate the novelty and advantage of our approach. We now provide, to our knowledge, the first morphological evidence of a re-established dermal plexus (lines 156-157, figure 2C, D) as well as morphological and very robust electrophysiological evidence of muscle spindle reinnervation via myelinated nerve fibers (lines 159-161, figure 3B).

No evidence was provided to suggest reinnervation of specific sensory organelles (Meissner's corpuscles, etc) within the glabrous skin graft. This weakens the case for using this relatively scarce skin type for as a graft vs non-glabrous skin grafting or the native skin overlying the target muscle.

RE: We acknowledge the lack of morphological evidence regarding specific mechanoreceptors in our original submission. We have therefore added such morphological analyses to the manuscript. The newly added data demonstrates specific Meissner corpuscle reinnervation and the re-establishment of a dermal plexus with Merkel discs at the apex of the dermis (lines 156-157, figures 2B, C, D)

Direct mechanical stimulation of the target muscle without skin graft (the only control group) demonstrated greater amplitude ENR signals on average than mechanical stimulation of a glabrous skin graft overlying the target muscle. This could be explained by the reinnervated spindles within the target muscle which were nicely demonstrated by the authors. This finding also raises the possibility that the skin graft itself did not contribute to mechano sensitive signal transduction and may have actually provided a shielding effect of the mechanosensitive muscle itself. Again, the authors did not demonstrate reinnervation of mechanosensitive organelles in the skin grafts.

RE: *We further investigated the discrepancy between the amplitudes of the two groups. Our newly added morphological evidence suggests that transmuscular (i.e., the transferred sensory fibers entering and traversing the muscle and reaching the skin graft through the musculocutaneous junction) sensory reinnervation of the skin graft involves many unmyelinated fibers, while muscle spindles were reinnervated by large, myelinated fibers (lines 157-161, figure 3A, B). To our knowledge, this is the first data shedding light on the basic neurophysiological processes occurring during targeted sensory reinnervation. The literature on peripheral nerve recordings suggests that it is difficult to record unmyelinated fibers with extraneural methods (such as our hook electrode setup) and that more invasive methods such as microneurography (3) may be necessary. Considering the newly added evidence for mechanoreceptor reinnervation, this seems to be the most plausible explanation as to why we were not able to decipher afferent signals from the skin in the recorded compound signal. The skin may then indeed act as a mechanical shield, thus leading to slightly lower amplitudes in this group. It is currently unclear why myelinated afferent fibers are predominantly found with proprioceptive receptors and not in the skin following reinnervation. A possible explanation may be that the musculocutaneous junction simply acts as a mechanical barrier for myelinated fibers. However, this poses the question how patients who underwent TSR perceive tactile sensation in such a normal way as it is plausible to assume that this also occurred in clinical cases reported in the literature (4, 5). We suspect this may be in part due to processes at the level of the somatosensory cortex, however, this is far beyond the scope of this manuscript and purely speculative.*

Reviewer #2 (Remarks to the Author):

The manuscript by Festin and colleagues reports on the creation of a novel biological sensorimotor interface for potential bionic reconstruction in a rat model. Overall, the rationale behind the paper is sound, and the results are encouraging.

RE: *Thank you for the positive feedback.*

However, there are several limitations which decrease enthusiasm for the paper. The authors state that their model is novel, however, it is not very different from other models already published (i.e., VDMT, C-RPNI). The authors do acknowledge this, but it limits overall enthusiasm for the novelty of the construct.

RE: *The novelty of our approach lies in the advancement of TSR. This includes the reinnervation of a transplanted glabrous skin graft as opposed to non-glabrous, overlying skin as well as demonstrating the possibility of restoring proprioceptive feedback via reinnervated muscle spindles. We have improved the discussion of these aspects in the revised manuscript with the inclusion of new analyses and results, as it is described in the previous and following replies.*

Specifically, their construct is very similar to VDMT since they take an intact muscle and denervate it, but they leave the vascular pedicle and also leave the muscle insertion to the bone intact. It is not clear what the construct even is until you read the methods section. It was unclear whether it was an

intact muscle with intact neurovascular structures or a muscle graft or some other surgical technique.

RE: *The concept of TMR and TSR (or simply targeted reinnervation as a combined term) was described in the introduction (lines 82-96). The approach was also described as a targeted reinnervation model (line 108) with a brief description of the surgical approach. We acknowledge the ambiguity regarding the targeted muscle in the introduction section and added further description (line 110). While it is true that the VDMT concept is anatomically similar, it is a concept that was described much later than targeted reinnervation and is used for treating neuromas and not as a human-machine interface.*

The primary concept of the paper is a "bidirectional" interface. However, the results do not discuss anything about the ability of the interface to facilitate motor control.

RE: *The efferent aspect has been investigated by our group in great detail in previous work (6). This trial was built upon the previous one, therefore it was not justifiable to reproduce these results both from an ethical and resource point of view.*

On the flip side, while they discuss electrophysiological sensory results, there is no immunohistochemistry showing reinnervation of sensory end organ targets in the skin. The figures showing "reinnervation of the skin" are not showing that the fibers are coming from the muscle side. If the skin is really reinnervated by the transferred nerve, it would show as fibers coming from the muscle into the skin.

RE: *We have now added immunohistochemical evidence of Meissner corpuscle reinnervation. Furthermore, the new morphological evidence also suggests the re-establishment of a dermal plexus with Merkel discs at the apex of the dermis (lines 156-157, figure 2B, C, D). The evidence already present in the original version of the manuscript clearly demonstrates sensory nerve fibers extending from the muscle through the musculocutaneous junction into the graft (figure 5). Furthermore, we now provide more immunohistochemical evidence of nerve fiber bundles crossing the musculocutaneous junction (figure 2A).*

There are no postoperative photos of the construct showing that there is survival of the denervated muscle and the overlying skin graft. If there are really a lot of cysts like they said in the text, from a clinical perspective, wouldn't a secondary surgery be needed to remove all the cysts and wouldn't this cause a lot of pain given c-fiber transection during cyst removal?

RE: *A picture of the biological interface demonstrating its survival has now been added to the supplemental material (supplemental figure S2A). Despite microsurgically removing the epithelium of the graft, a minimal amount must have remained thus leading to cysts. In case of this trial, it seems to be a matter of size. The rat skin graft is very thin and delicate, which is why more aggressive removal is not possible due to potentially destroying the graft. We do not think that this will be of major concern in clinical practice as de-epithelization is a standard procedure in plastic surgery.*

And another surgery for electrode implantation once this heals?

RE: *We are, in fact, investigating on how to interact with this interface, however, at this point we are not able to delineate a final surgical approach in a clinical setting. We merely hypothesized whether a magnetic communication interface may be feasible (lines 396-402) and what a "bio-hub" of the future may look like (lines 413-417).*

There is an issue with deciphering afferent activity of the sensorimotor interface. The authors used a construct without skin graft (so muscle only) as their control. But this control construct had robust

signals to light touch and vibration. The authors concluded that the response to the monofilament and vibration was proprioceptive in nature which doesn't make sense as muscle spindles are not directly involved with responding to fine touch sensations like those detected by cutaneous mechanoreceptors.

RE: *It is correct that muscle spindles are not involved in the perception of light touch. However, the monofilaments cause a mechanical displacement of the muscle (i.e. stretch). Muscle spindles react both statically and dynamically to changes in length (7) which is the most plausible explanation for our electrophysiological data. This also applies for vibration as it constantly induces small changes in muscle length. Data has demonstrated that muscle spindles respond to vibration of up to 180Hz (8), while we used approximately 100Hz.*

How do they know that instead, it isn't a result of direct activation of the nerve? How does this compare to normal intact innervated skin or normal intact innervated muscle? Furthermore, there are no stats provided comparing to the control. The idea of the construct having proprioceptive signaling abilities is a strong point of the paper. However, pulling on the muscle will inevitably pull on the nerve and thus create a movement artifact. How are the authors certain that the readings they are getting are actually proprioception in nature and not simply movement artifact?

RE: *Movement artifacts are definitely a confounder to be concerned about. For this reason, we applied the vibration stimulus (as it was the mechanically strongest) within 1mm of the transferred nerve while recording the afferent nerve signals. We have now added the recorded ENG signals following perineural stimulation which verify that our data is not the results of movement artifacts (lines 234-240, supplemental table S4).*

There are some issues with the retrograde labeling results. First, it is very hard to see the RB's in Figure 6. It is unclear as to why the authors didn't use a secondary tracer like FluoroRuby, which would have also stained the entire cell body like FB. In the current study, how do the authors know that the RB labeled DR cells are all from nerves reinnervating the dermal graft? The skin graft is very thin and it is definitely possible for the dye to be picked up by the muscle and thus it could be from afferent nerves with muscle origin.

RE: *This is an important point. FluoroRuby was initially used for intradermal injection, but we noticed that it seeped into the muscle, thus being picked up by intramuscular nerve fibers (both motor and sensory). This led to unspecific labeling in the DRG and also labeled motoneurons in the spinal cord. This is why we opted for Retrobeads. They are more difficult to evaluate as one has to count the small, intracellular spheres, but it is possible, as also demonstrated by other authors (9), and they do not seep into the underlying muscle, but basically stay where they are injected (10). We established a method for exact intradermal application during the pilot trials using a delicate Hamilton syringe and confirmed this by evaluating the spinal cord. As can be seen in our results, we had virtually no uptake of Retrobeads into the spinal cord (lines 194-196) which would be expected if the tracer were picked up intramuscularly.*

1. Kim, S.H. and Y.H. Lee, *Re-evaluation of the distribution of Meissner's corpuscles in human skin*. *Anat Cell Biol.* **53**(3): p. 325-329 (2020).
2. Svientek, S.R., D.C. Ursu, P.S. Cederna, and S.W.P. Kemp, *Fabrication of the Composite Regenerative Peripheral Nerve Interface (C-RPNI) in the Adult Rat*. *J Vis Exp*(156)(2020).
3. Raspopovic, S., A. Cimolato, A. Panarese, F. Vallone, J. Del Valle, S. Micera, and X. Navarro, *Neural signal recording and processing in somatic neuroprosthetic applications. A review*. *J Neurosci Methods.* **337**: p. 108653 (2020).
4. Kuiken, T.A., P.D. Marasco, B.A. Lock, R.N. Harden, and J.P. Dewald, *Redirection of cutaneous sensation from the hand to the chest skin of human amputees with targeted reinnervation*. *Proc Natl Acad Sci U S A.* **104**(50): p. 20061-6 (2007).
5. Marasco, P.D., A.E. Schultz, and T.A. Kuiken, *Sensory capacity of reinnervated skin after redirection of amputated upper limb nerves to the chest*. *Brain.* **132**(Pt 6): p. 1441-8 (2009).
6. Bergmeister, K.D., M. Aman, S. Muceli, I. Vujaklija, K. Manzano-Szalai, E. Unger, R.A. Byrne, C. Scheinecker, O. Riedl, S. Salminger, F. Frommlet, G.H. Borschel, D. Farina, and O.C. Aszmann, *Peripheral nerve transfers change target muscle structure and function*. *Sci Adv.* **5**(1): p. eaau2956 (2019).
7. Kröger, S. and B. Watkins, *Muscle spindle function in healthy and diseased muscle*. *Skelet Muscle.* **11**(1): p. 3 (2021).
8. Roll, J.P., J.P. Vedel, and E. Ribot, *Alteration of proprioceptive messages induced by tendon vibration in man: a microneurographic study*. *Experimental Brain Research.* **76**(1): p. 213-222 (1989).
9. da Silva Serra, I., Z. Husson, J.D. Bartlett, and E.S. Smith, *Characterization of cutaneous and articular sensory neurons*. *Mol Pain.* **12**(2016).
10. Lumafluor. *Protocol*. [cited 2022 March 16]; Available from: <https://lumafluor.com/protocol> (2015).

REVIEWERS' COMMENTS

Reviewer #1 (Remarks to the Author):

The authors have added new data that improved the significance and novelty of this work.

Reviewer #2 (Remarks to the Author):

The authors have addressed the majority of questions presented by both reviewers, and they have substantially added to the manuscript. I have no further comments/edits for the paper.